# Quantum criticality at the superconductor-insulator transition revealed by specific heat measurements

S. Poran[1,2], T. Nguyen-Duc[2,3], A. Auerbach[4,5], N. Dupuis[5], A. Frydman[1,2,3] & Olivier Bourgeois[2,3]

The superconductor–insulator transition (SIT) is considered an excellent example of a quantum phase transition that is driven by quantum fluctuations at zero temperature. The quantum critical point is characterized by a diverging correlation length and a vanishing energy scale. Low-energy fluctuations near quantum criticality may be experimentally detected by specific heat, $c_p$, measurements. Here we use a unique highly sensitive experiment to measure $c_p$ of two-dimensional granular Pb films through the SIT. The specific heat shows the usual jump at the mean field superconducting transition temperature $T_c^{mf}$ marking the onset of Cooper pairs formation. As the film thickness is tuned towards the SIT, $T_c^{mf}$ is relatively unchanged, while the magnitude of the jump and low-temperature specific heat increase significantly. This behaviour is taken as the thermodynamic fingerprint of quantum criticality in the vicinity of a quantum phase transition.

[1] Department of Physics, Bar Ilan University, Ramat Gan 52900, Israel. [2] Institut NÉEL, CNRS, 25 avenue des Martyrs, F-38042 Grenoble, France. [3] Univ. Grenoble Alpes, Inst NEEL, F-38042 Grenoble, France. [4] Department of Physics, Technion, 32000 Haifa, Israel. [5] Laboratoire de Physique Théorique de la Matière Condensée, CNRS UMR 7600, UPMC-Sorbonne Universités, 4 Place Jussieu, 75252 Paris, France. Correspondence and requests for materials should be addressed to A.F. (email: aviad.frydman@gmail.com) or to O.B. (email: olivier.bourgeois@neel.cnrs.fr).

Quantum criticality is a central paradigm in physics. It unifies the description of diverse systems in the vicinity of a second-order, zero-temperature quantum phase transition, governed by a quantum critical point (QCP). QCPs have been discovered and extensively studied primarily in metallic and magnetic systems. At many of these QCPs, especially in two dimensions (2D), conventional mean-field and Fermi-liquid theories fail in lack of well-defined quasiparticles. QCPs inspired innovative non-perturbative approaches[1–4], including those relevant to field theory/gravity duality[5].

In 2D superconducting films, the zero-temperature superconductor–insulator transition (SIT) has been viewed as a prototype of a quantum phase transition that is controlled by a non-thermal tuning parameter $g$[6]. Experimentally, the transition has been driven utilizing various $g$ such as inverse thickness[7–16], magnetic field[12,13,17–26], disorder[25,27,28] chemical composition[29] and gate voltage[30]. For $g < g_c$ the film is a superconductor with well-defined quasiparticles and superconducting collective modes as well as a finite 2D superfluid density. As $g$ is increased, the system enters the critical regime in which excitations are strongly correlated, while the superfluid density vanishes as $g \to g_c$. For $g > g_c$, the system becomes insulating with gapped charge excitations.

2D superconducting granular films have been shown to exhibit signs for Cooper-pairing effects such as the presence of an energy gap, $\Delta$ well into the insulator phase[31], and it has been argued that they may be modelled by a bosonic quantum field theory with O(2) symmetry[32]. Similar findings were found for disordered thin films[27,29,33] in which disorder is assumed to generate emergent granularity[34]. If one ignores the broadening of the SIT due to inhomogeneities, divergent correlation length and time are expected at the transition. Indeed, recent optical conductivity measurements have detected signatures of the critical amplitude (Higgs) mode, becoming soft at the SIT[35,36].

One of the salient thermodynamic signatures of a QCP is the presence of excess entropy or specific heat[1,2]. In heavy electron metals, this has been observed as the divergence of linear specific heat coefficients, or electronic effective mass, which signals a deviation from conventional Fermi-liquid theory behaviour[37]. Such a signature would be very important to measure in the SIT system, to probe the critical thermodynamics near this non-mean-field-type QCP.

Entropy $S(T)$ is a fundamental physical quantity of significant importance for the quantum phase transition; however, its absolute value cannot be directly measured. On the other hand, the specific heat $c_p(T)$ at constant pressure can have its absolute value determined experimentally. Apart from its interest regarding QCP, it is above all the most appropriate physical property to categorize the order of a phase transition or to probe signatures of fluctuations. Despite its crucial interest, specific heat experiments have never been performed close to the QCP in the context of the SIT. The main reason is that the systems under study are 2D ultra-thin films involving ultra-low mass. The substrate mass onto which thin films are deposited is usually much larger than that of the film itself rendering ultra-thin film specific heat unmeasurable.

Here we report on the first experimental demonstration of excess specific heat in 2D granular films spanning the superconductor–insulator quantum phase transition. We employ a unique technique based on thin suspended silicon membrane used as a thermal sensor that enables ultra-sensitive measurements of the specific heat of superconducting films close to the SIT. We find that the mean-field critical temperature, $T_c^{mf}$, remains basically unchanged through the SIT. Nevertheless, the specific heat jump at $T_c^{mf}$ and the specific heat magnitude at temperatures lower than the critical temperature increase progressively towards the transition. These results are interpreted as thermodynamic indications for quantum criticality close to the QCP.

## Results

**Specific heat experimental set-up.** The samples used in this study were ultra-thin granular films grown by the 'quench condensation' technique. In this method, sequential layers are deposited directly on an insulating substrate held at cryogenic temperatures ($T = 8$ K) under ultra-high-vacuum conditions[7,38–40]. The first stages of evaporation result in a discontinuous layer of isolated superconducting islands. As material is added, the inter-grain coupling increases and the system undergoes a transition from an insulator to a superconductor.

The measurement set-up consists of a thermally sensitive thin membrane that is suspended by 10 silicon arms used for mechanical support, thermal isolation and for electrical wiring (Fig. 1). This results in a calorimetric cell that enables the simultaneous measurement of transport properties and heat capacity with energy sensitivity as low as an attoJoule around 1 K (ref. 41) so that temperature variation as low as few microkelvin can be detected on ultra-thin samples with masses down to few tens of nanograms. This set-up provides a unique opportunity to measure simultaneously the film resistance, $R$, and heat capacity, $C_p$, of a single film as a function of thickness through the entire SIT without the need to warm up the sample or to expose it to atmosphere; both processes being harmful to ultra-thin films. Further experimental details are specified elsewhere[42], see also Methods.

**Specific heat and resistance through the SIT.** Panels a and b of Fig. 2 show concomitant $R(T)$ and $C_p(T)$ measurements performed on a series of 18 consecutive depositions of Pb on a single nano-calorimetric cell. The thinnest layer is an insulator with $R \gg 1 \, G\Omega$, making it unmeasurable within the sensitivity of our set-up, and the thickest is a superconductor with a sharp transition corresponding to the bulk critical temperature of Pb. The $R(T)$ curves are typical of granular Pb films[11,31], where the first six depositions are on the insulating side of the SIT, the evaporations 7 and 8 show resistance re-entrance behaviour, 9–16 are superconducting with long exponential tails that become increasingly sharper until, in stages 17 and 18, the transition is sharp.

Such samples have been considered as prototype systems for the bosonic SIT in which the grains are believed to be large enough to sustain superconductivity with bulk properties[11,31,39,43]. However, for the thinnest layers, phase fluctuations between the grains are strong enough to suppress global superconductivity and lead to an insulating state. The critical temperature measured by transport (temperature of zero resistance) is thus governed by order-parameter fluctuations and not by actual pair breaking.

As opposed to electrical transport, thermodynamic measurements can be performed deep into the insulating regime (purple line in Fig. 2b). The measured heat capacity contains contributions from phononic, electronic and superconducting degrees of freedom. In the following, we will only focus on the specific heat $c_p$ (the heat capacity $c_p$ normalized to the mass). Above $T_c$, the normal specific heat, $c_n$, should follow the well-known form:

$$\frac{c_n}{T} = \gamma_n + \beta T^2 \qquad (1)$$

with $\gamma_n$ and $\beta$ proportional to the electron and phonon specific heat (heat capacity divided by the mass of the Pb film), respectively. In our case, the phonon contributions to $c_p$

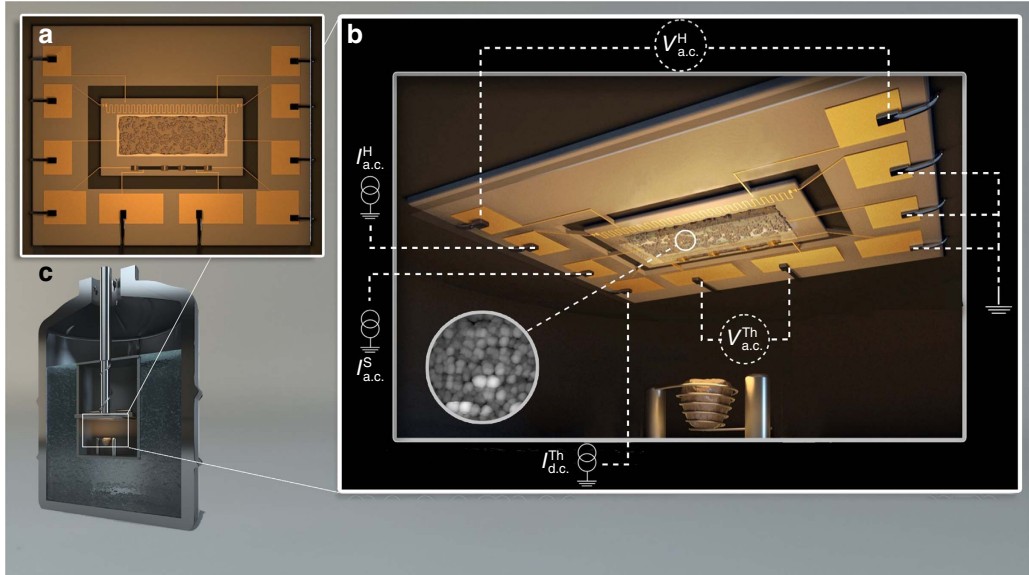

**Figure 1 | Sketch of the experimental set-up. (a)** The suspended membrane acting as the thermal cell contains a copper meander, used as a heater, and a niobium nitride strip, used as a thermometer. These are lithographically fabricated close to the two edges of the active sensor. **(b)** The quench condensation set-up is constituted by an evaporation basket containing the Pb material that is thermally evaporated on the substrate held at cryogenic temperatures and in UHV conditions. The granular quench-condensed film is evaporated through a shadow mask which, together with the measurement leads, defines its geometry. The biasing of the heater is done with a a.c. current $I_{a.c.}^H$ (used for heat dissipation), $I_{d.c.}^{Th}$ is the d.c. current biasing the thermometer for the measurement of the temperature through the voltage $V_{a.c.}$ and the measurement of the resistance of the quench-condensed films is done using the d.c. current $I_{d.c.}^S$. The inset shows a low-temperature STM image of the quench-condensed granular Pb[56]. **(c)** The whole experimental set-up is immersed in a liquid helium bath.

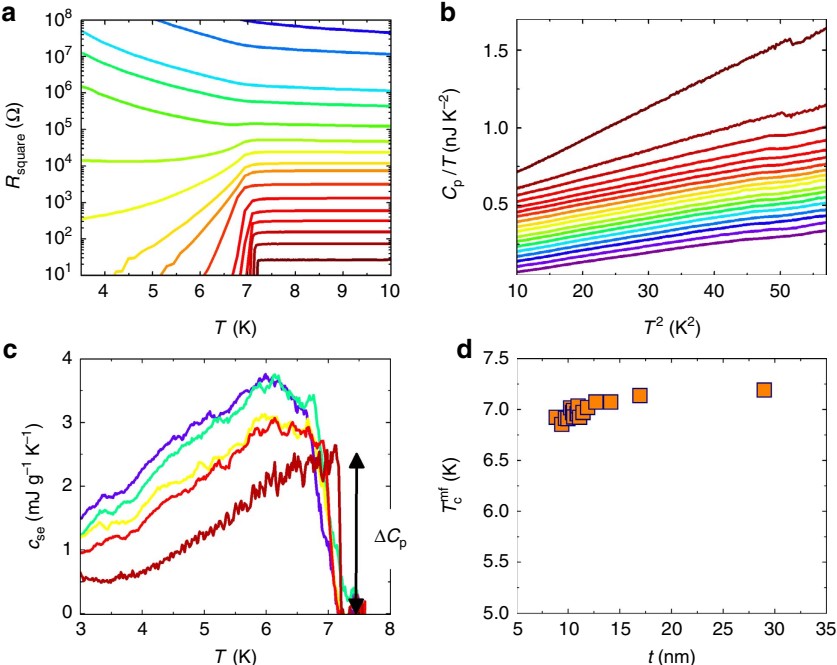

**Figure 2 | Resistance and heat capacity versus temperature. (a)** Resistance per square and **(b)** the heat capacity of ultra-thin lead versus temperature of the 18 sequential deposited films. The same colour code is used for all the panels of the figure. The resistances of the two first depositions are unmeasurable, unlike heat capacity that can be measured deeply in the insulating regime (in purple in **b**). **(c)** $c_{se}$ versus $T$ for a number of layers clearly depicting the growth of specific heat as the sample is thinned. The thicknesses are 8.9, 10.2, 10.95, 12.7 and 29 nm from top to bottom, respectively. **(d)** Quasi-constant mean-field critical temperature, $T_c^{mf}$ of the granular Pb layers as extracted from the midpoint of the heat capacity jump as a function of film thickness through the SIT.

overwhelms the electronic contribution by at least one order of magnitude as can be seen from the linear behaviour of the heat capacity shown in Fig. 2b. This strong phonon contribution originates from the amorphous nature of the superconducting Pb materials obtained by quench condensation; the discussion of this point goes far beyond the scope of this paper (see Methods). Hence, the specific heat above $T_c$ can be fit using a simpler relation than equation (1), that is, $\frac{c_n}{T} = \beta T^2$ since $\gamma_n \ll \beta T^2$.

In this study, we focus on $c_{se}$, the specific heat of electrons in the superconducting state compared with the one in the normal state, as defined by the following relation: $c_{se}(T) = c_p(T) - c_n(T)$. To do this, we subtract from each of the measured $c_p(T)$ of Fig. 2b the specific heat extrapolated from the normal-state regime $c_n(T) = \beta T^3$ above $T_c$. The superconducting electronic specific heat $c_{se}$ (J g$^{-1}$ K$^{-1}$) for a number of selected layers are shown in Fig. 2c. The remainder of this article will focus on $c_{se}$ of the films close to the quantum phase transition.

Two prominent facts are clearly observed in the specific heat data of Fig. 2c. The first is the position of the specific heat jumps $\Delta c_p$ and the second is the amplitude of the specific heat below the critical temperature.

**Specific heat jumps $\Delta c_p$.** The amplitude of these specific heat jumps at the transition temperature $T_c^{mf}$ is quantified by: $\Delta c_p = c_p(T = T_c^{mf}) - c_n(T = T_c^{mf}) = c_{se}(T = T_c^{mf})$. These jumps are identified as the usual fingerprint of the superconducting second-order phase transition.

The temperature of the jumps $T_c^{mf}$, (defined by the midpoint of the jumps) is close to the superconducting transition temperature of bulk Pb $T_c^{bulk} = 7.2$ K. However, $T_c^{mf}$ does not mark the onset of macroscopic superconductivity since the thinnest films remain resistive below it (Fig. 2a). It rather reflects the onset of local, intra-grain Cooper pairing. Interestingly, while the jump becomes increasingly broadened for thinner layers, $T_c^{mf}[t]$ changes very little with thickness $t$ (Fig. 2d). This is true in the whole regime between the thickest layer, which is a high-quality super-conductor, and the thinnest layer, which is insulating at low temperatures. In fact $T_c^{mf}$ has similar behaviour to the tunnelling gap $2\Delta^{tunn}$, which was found to remain unchanged through the SIT[31]. The fact that it does not decrease towards the insulating state means that the SIT is driven by inter-grain phase fluctuations. A model for such a bosonic SIT transition is provided by a disordered network of weakly Josephson-coupled, low-capacitance superconducting grains[44].

**Excess specific heat below $T_c^{mf}$.** The key observation of Fig. 2c is that the electronic specific heat in the superconducting state increases for thinner films. This increase is throughout the tempe-rature range $3 < T < T_c^{mf}$. The specific heat jump $\Delta c_p$ also increases as $t$ decreases towards the SIT as seen in Fig. 2c. Since superconducting grains are involved, one may naively expect that the anomaly at $T = T_c$ would be spread over a broader temperature range as the grains become smaller as calculated by Muhlschlegel[45,46], and hence reduced in amplitude. Indeed, superconductivity may be suppressed in very small grains due to energy level splitting being of the order of the superconducting gap and the specific heat anomaly is expected to be less pro-nounced. Here the opposite is observed: $\Delta c_p$ is more pronounced as the film is made thinner and pushed towards the QCP as illustrated in Fig. 3. We note that demonstrating a decrease of specific heat as the sample crosses the QCP is extremely difficult for two main reasons: first, pushing the sample into the insulating regime requires increasingly thinner films. This results in the signal-to-noise ratio becoming less and less favourable. Second, we cannot enter too deeply into the insulating regime because this

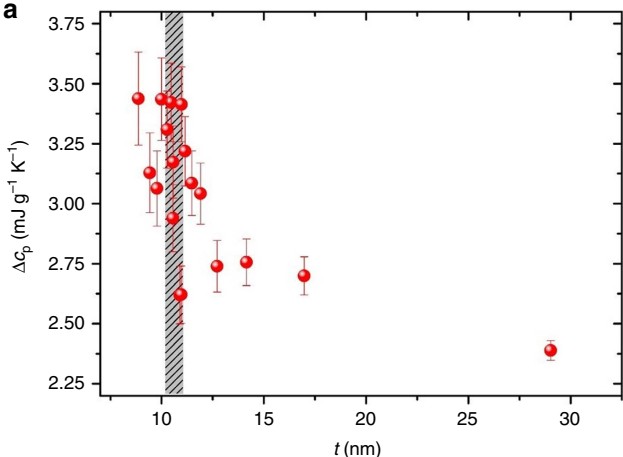

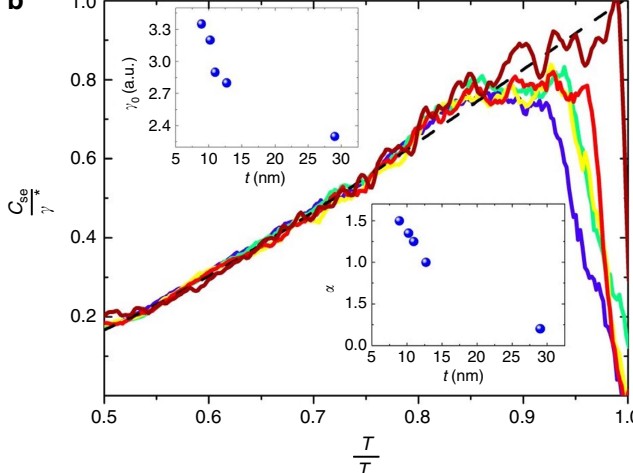

**Figure 3 | Excess specific heat. (a)** The specific heat jump, $\Delta c_p$, versus the thickness of the layer. The hatched region marks the position of the QCP. The error bars are estimated from the noise of the heat capacitance measurements which becomes larger the thinner the film. **(b)** The superconducting electronic specific heat, $c_{se}$ for the layers of Fig. 2c scaled according to equations (6 and 7). The colour code is similar to that of Fig. 2c. The insets show that both $\alpha(t)$, (the power of equation (7)) and $\gamma_n$ increase towards the QCP. The dashed black line is the BCS prediction of equation (3).

would require measurements of a very-low-mass sample beyond the sensitivity of our set-up (which is state-of-the-art for specific heat experiments).

**Discussion**

The results presented show that the low-temperature specific heat is enhanced towards the QCP. Here we present a possible scenario for the effect of quantum criticality on the electronic specific in the superconducting state. We recall that for a weakly interacting metal, such as Pb, the normal-state specific heat follows equation (1), where the electronic contribution coefficient is given by:

$$\gamma_n = \frac{\pi^2}{3} k_B^2 g(\epsilon_F) \qquad (2)$$

here $g(\epsilon_F)$ is the single-particle density of states at the Fermi energy. Hence, for free electrons, $\gamma_n = m\frac{k_B^2 k_F}{3\hbar^2}$ is proportional to the electron mass.

Excess specific heat is not expected near an ordinary disorder driven metal-to-insulator transition via Anderson localization. In such a transition, the linear coefficient persists into the insulator phase as measured in, for example, silicon-doped phosphorous system[47]. Therefore the enhancement observed here at low temperatures, $T < T_c^{mf}$, indicates that it is related to pairing and superconductivity.

The BCS prediction for the electronic specific heat in the superconducting state is given by:[48]

$$c_s(T) \simeq 10\gamma_n T_c \exp\left(-1.76\frac{T_c}{T}\right) \quad (3)$$

hence, it depends on the same $\gamma_n$ as that of the normal state. In addition, the BCS specific heat jump also scales with $\gamma_n$ since[48],

$$\frac{\Delta c_p}{c_n} \simeq 1.43 \quad (4)$$

Since, as we demonstrated experimentally, $T_c^{mf}$ remains constant through the SIT, we interpret the $c_s$ enhancement as a signature of the renormalization of the electron mass appearing in the coefficient $\gamma_n$. This is described through the presence of a self-energy in the Green function formalism, which is an interaction driven effect in the vicinity of the QCP. Indeed close to a QCP, following quantum field theory, the electron effective mass is renormalized by the self-energy $\Sigma$ by[49,50]:

$$\frac{\gamma^*}{\gamma_n} = \frac{1 + \partial_{\epsilon_k}\Sigma(\epsilon_{k_F}, \omega)}{1 - \partial_\omega\Sigma(\epsilon_{k_F}, \omega)} \quad (5)$$

where $\Sigma$ depends on the many-body interactions. These interactions can be with phonons or with other electrons. However, the main contribution to the self-energy arises from the interaction of the fermions with low-energy collective superconducting quantum fluctuations. These quantum fluctuations can be either gapless phase-density fluctuations (called Goldstone/plasmon modes) or amplitude fluctuations (called Higgs modes), both lead to an infrared singularity in the self-energy[51]. This can be phenomenologically modelled by a divergent $\gamma^*(T)$ by replacing equation (3) by:

$$c_s(T, t) \simeq 10 T_c^{mf}\gamma^*(T, t)\exp\left(-1.76\frac{T_c^{mf}}{T}\right) \quad (6)$$

The specific heat curves depicted in Fig. 2c imply that such an interpretation would require a $\gamma^*(T)$ that would significantly increase at low temperatures. We note that a divergence of linear specific heat coefficient $\gamma_n$ has been widely observed at the QCP separating a magnetic and paramagnetic Fermi liquids[2] in good agreement with what is currently observed in granular superconducting Pb. In those systems, the temperature dependence of $\gamma_n$ was predicted to be[2]: $\gamma_n(T) \sim \gamma_0 \log(1/T)$. In our limited temperature interval, we find that a better fit is given by assuming a power-law dependence of $\gamma^*$ on temperature (Fig. 3):

$$\gamma^* = \gamma_0 \left[T_c^{mf}/T\right]^\alpha, \quad (7)$$

where both $\gamma_0$ and $\alpha$ increase as the thickness $t$ decreases towards the QCP. The appropriateness of such scaling is illustrated in Fig. 3, where all curves can be collapsed by the same equation 7.

Finally, the bosonic degrees of freedom due to order-parameter fluctuations of phase and amplitude should also contribute to the specific heat. Indeed, such specific heat, $c^{boson}$, was computed for the 2D O(2) relativistic Ginzburg–Landau theory, by the non-perturbative renormalization group[4]. The results are shown in the Methods section. $c^{boson}/T^2$ exhibits a peak that reflects a relative enhancement of order 2 at the QCP. This peak is associated with the excess of low-temperature entropy of the softening amplitude fluctuations (Higgs mode). As for the granular superconducting

films, the overall magnitude of the bosonic specific heat is controlled by the density of grains so that $c^{boson} \sim n_{grains} k_B$, which is at least two orders of magnitude smaller than the measured excess specific heat scale. Hence, the effects of the bosonic collective modes close to the QCP are measurable only indirectly through their effects on the electronic effective mass.

To conclude, we experimentally demonstrated the enhancement of specific heat towards the QCP in a granular superconductor. This is interpreted as the thermodynamic indication for quantum criticality in the quantum critical regime of the SIT. A possible mechanism for this specific heat enhancement is the increase of the electronic effective mass in the vicinity of the quantum phase transition. The effective mass increase is correlated to the self-energy emerging from the interactions between the fermions and bosonic collective modes that become pronounced close to the quantum critical point of the SIT. From our results, it is not possible to tell whether the specific heat decreases as the sample is pushed deep into the insulating phase. The data become too noisy to draw any definite conclusion since the film becomes too thin. The direct detection of the collective modes by specific heat require either more sensitive instrumentation or more adapted systems.

## Methods

**Experimental methods.** The samples used in this study are ultra-thin granular films quench condensed directly on an insulating substrate. In these systems, superconducting layers are sequentially evaporated on a cryogenic substrate under ultra high vacuum (UHV) conditions[38]. The first stages of evaporation result in a discontinuous layer of isolated superconducting islands. As material is added the grains become larger and thereby the inter-grain coupling increases.

The experimental set-up is composed of a thin silicon membrane on which the evaporation of Pb is performed. A copper meander to be used as a heater and a niobium nitride strip to be used as a thermometer, both close to two edges of the active sensor, are structured by photolithography and lift-off processes. The sensitive part of the membrane is suspended by 10 silicon arms holding the electrical wiring. This results in a calorimetric cell into which one can supply heating power and measure its temperature while being effectively separated from the heat bath. Transport measurements of the thin Pb films was enabled by depositing 5 nm titanium and 25 nm gold on two additional leads through a mechanical mask. The quench condensation apparatus consists of a high-vacuum chamber containing tungsten thermal evaporation boat. The membrane is wire-bonded to the sample holder that is mounted in the quench condensation chamber. Quench condensation evaporations are made through a mechanical shadow mask defining a window of 1.3 mm × 3.2 mm on the membrane. The chamber is then immersed in liquid helium, and by pumping on a 1-K pot the system is capable of reaching 1.5 K.

Measurement of the membrane heat capacity is conducted by a.c. calorimetry, in which a current at frequency $f$ is driven through the heater. This oscillates the cell temperature at the second harmonic $2f$ with amplitude $\delta T_{a.c.}$. This amplitude is picked up by the thermometer, which allows the calculation of heat capacity through:

$$C_p = \frac{P_{a.c.}}{4\pi f \delta T_{a.c.}} \quad (8)$$

$P_{a.c.}$ being the joule heating power dissipated in the heater, allowing highly sensitive

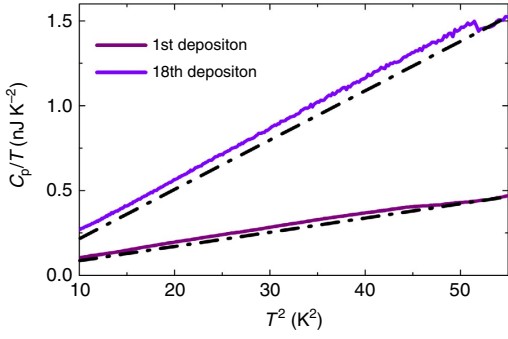

**Figure 4 | Heat capacity of the granular Pb versus $T^2$.** The linear cubic fit in temperature is used to extract the electronic contribution to the specific heat in the superconducting state $c_{se}$ without the phononic contribution.

**Table 1 | Experimental data extracted from the heat capacity measurement of the 18 evaporations (refereed to by sample no).**

| Deposition no. | Mass (μg) | $C_p$ (nJ K$^{-1}$) | $\beta$ (μJ K$^{-4}$) | $t$ (nm) | $\Delta C_p$ (nJ K$^{-1}$) | $\Delta c_p$ (mJ g$^{-1}$ K$^{-1}$) | $R_{sq}$ (Ω) | $T_c$ (K) |
|---|---|---|---|---|---|---|---|---|
| 1 | 0.42 | 16 | 97 | 8.9 | 1.44 | 3.5 | NA | 7.02 |
| 2 | 0.44 | 17 | 96 | 9.4 | 1.39 | 3.1 | $6.6 \times 10^8$ | 7.02 |
| 3 | 0.46 | 18 | 98 | 9.7 | 1.41 | 3.05 | $7.8 \times 10^7$ | 7.08 |
| 4 | 0.47 | 18.3 | 98.6 | 10 | 1.62 | 3.45 | $1.5 \times 10^7$ | 7.2 |
| 5 | 0.48 | 18.8 | 98.5 | 10.2 | 1.60 | 3.3 | $1.5 \times 10^6$ | 7.2 |
| 6 | 0.49 | 19.2 | 98.5 | 10.4 | 1.68 | 3.4 | $0.5 \times 10^6$ | 7.13 |
| 7 | 0.497 | 19.3 | 97.4 | 10.5 | 1.46 | 2.95 | $1.5 \times 10^5$ | 7.12 |
| 8 | 0.498 | 19.4 | 99 | 10.6 | 1.58 | 3.2 | $4.8 \times 10^4$ | 7.15 |
| 9 | 0.514 | 19.9 | 97 | 10.85 | 1.35 | 2.6 | $2.4 \times 10^4$ | 7.12 |
| 10 | 0.517 | 20 | 97 | 10.95 | 1.35 | 2.6 | $1.1 \times 10^4$ | 7.1 |
| 11 | 0.518 | 20.1 | 98 | 11 | 1.76 | 3.4 | 7,160 | 7.2 |
| 12 | 0.526 | 20.4 | 97.7 | 11.1 | 1.69 | 3.2 | 3,080 | 7.2 |
| 13 | 0.54 | 21 | 98 | 11.4 | 1.67 | 3.1 | 1,380 | 7.02 |
| 14 | 0.56 | 22.4 | 99.5 | 11.9 | 1.71 | 3.05 | 647 | 7.17 |
| 15 | 0.6 | 23 | 98 | 12.7 | 1.65 | 2.75 | 330 | 7.08 |
| 16 | 0.67 | 26 | 98.5 | 14.1 | 1.85 | 2.75 | 168 | 7.2 |
| 17 | 0.8 | 31 | 99.5 | 17 | 2.15 | 2.7 | 75 | 7.12 |
| 18 | 1.37 | 53 | 100 | 29 | 3.25 | 2.4 | 27 | 7.18 |

For each evaporation of Pb, we give the mass, the heat capacity $C_p$ at 7.5 K, the $\beta_i$ used in the equation $C_i(T) = \beta_i T^3$ for the fit, the thickness $t$, the heat capacity jump $\Delta C_p$ at $T_c$, the specific heat jump $\Delta c_p$ at $T_c$, the resistance per square $R_{sq}$ and the $T_c$ extracted from the heat capacity measurements.

heat capacity measurement[41]. To extract only the heat capacity of the evaporated metallic layer, the heat capacity of the raw membrane (containing the heater and the thermometer) is measured from 1.5 to 10 K. This background is subtracted from all the measurements we report in this letter. Further details on the experimental system can be found elsewhere[42].

**Specific heat components.** The specific heat (the heat capacity normalized to the mass of the sample) of the granular film has two contributions

$$c_n(T) = \gamma_n T + \beta T^3, \qquad (9)$$

one linear in temperature that comes from the electronic part of the degrees of freedom, and the second part cubic in temperature, coming from the phonons of the lattice. In our samples, as an experimental fact, the phonon contribution is far bigger than its electronic counterpart: $\gamma_n T \ll \beta_i T^3$; this is illustrated in Fig. 4 by the dominating cubic variation of the heat capacity versus temperature. Such behaviour has been already observed in many granular systems such as granular Al or granular Al-Ge. The large phonon contribution to specific heat may come from a lower Debye temperature in the thin film than in the bulk, additional degrees of freedom from surface phonons (soft surface modes) or amorphous structure of the quench-condensed Pb grains[52,53].

To extract information on the electronic specific heat in the superconducting state, one has to remove the contribution of phonons from the overall specific heat. In regular superconductors (crystals or thick films), the electronic specific heat dominates the total signal. Traditionally, to single out the superconducting electronic contribution to the specific heat, one can apply a magnetic field larger than the critical magnetic field, thus suppressing the superconducting state. In this limit, only the normal-state electrons contribute to the specific heat; the electronic contribution in the superconducting state $c_{se}$ is then obtained by subtracting the normal-state specific heat $c_n$ from the superconducting specific heat $c_s$: that is, $c_{se} = c_s - c_n$. Here this protocol cannot be applied since the expected critical magnetic field for small superconducting grains is far larger than the field that can be applied in our experiment ($B_c \gg 10$ T)[54]. If the critical magnetic field happens to be too high, there is a second traditional way to extract the electronic contribution, that is, by fitting the specific heat in the normal state above $T_c$ using equation (9), and then extrapolating to temperatures lower than $T_c$ to find $\gamma_n$. Again, this cannot be applied in our case since the electronic contribution in the normal state is overwhelmed by the phononic contribution. The consequence of this is twofold: (i) one cannot have access to $\gamma_n$ from the normal-state specific heat and (ii) one can only fit the normal state $c_n$ by a cubic power law.

Hence, to extract the significant information contained in the superconducting specific heat in granular materials, the following protocol has been used: the subtraction of the phononic contribution to the specific heat in the normal state is done by fitting the part of the specific heat curves above $T_c$ only by a cubic term with temperature such as $c_i(T) = \beta_i T^3$, $i$ being the $i$th evaporation. All the relevant numerical data extracted from the heat capacity measurements are gathered in the Table 1.

**Analysis of the bosonic contribution.** In this section, we present the analysis of the contribution to specific heat of quantum fluctuations within the bosonic model. Since the SIT is bosonic in nature, its universal properties can be described by a 2D

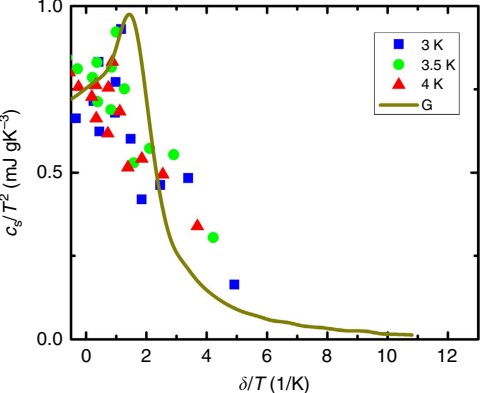

**Figure 5 | Scaling of the electronic specific heat.** The specific heat $c_s$ normalized to $T^2$ is scaled as a function of the characteristic energy scale $\Delta$ related to disorder normalized to temperature. The brown curve corresponds to the best adjustment obtained from the bosonic model developed by Rançon et al.[3] (see text).

effective bosonic theory. Standard scaling arguments imply that the singular part of the specific heat reads:

$$c_s = k_B \left( \frac{k_B T}{\hbar v_c} \right)^2 \mathcal{G}\left( \frac{\Delta}{k_B T} \right), \qquad (10)$$

where $G$ is a dimensionless scaling function. $v_c$ is the velocity of critical fluctuations at the QCP and $\Delta$ denotes a characteristic energy scale that vanishes at the QCP: $\Delta \propto |\delta - \delta_c|^{\nu z}$ with $\delta$ the non-thermal parameter that controls the transition (here the inverse of the film thickness), and $\nu$ and $z$ being the correlation length and the dynamical critical exponents, respectively. $\Delta$ corresponds to the excitation gap in the insulating phase and is defined by the superfluid stiffness in the superconducting phase.

The universal scaling function $\mathcal{F}$ defining the pressure $P(T) = P(0) + (k_B T)^3/(\hbar v_c)^2 \mathcal{F}(\Delta / k_B T)$ near the QCP has recently been calculated in the framework of the relativistic quantum O(2) model (quantum $\varphi^4$ theory for a complex field $\varphi$)[3]. The scaling function $\mathcal{G}(x)$ in equation (10) is simply $\mathcal{G} = 6\mathcal{F} - 4x\mathcal{F}' + x^2\mathcal{F}''$ (Fig. 5). A striking observation is that both the entropy and the specific heat are non-monotonic when $\delta$ is varied at fixed temperature, with a pronounced maximum near the QCP $\delta = \delta_c$. Although this result was obtained for a clean system, we expect it to hold also for a disordered system similar to the calculations of the collective amplitude modes in this region[55]. In any case, this scaling cannot explain the full physics of the data since at least two orders of magnitude differs between the expected specific heat variation in the bosonic picture and what has been observed on the granular superconducting films.

**Data availability.** The data that support the findings of this study are available from the corresponding author on reasonable request.

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

## Acknowledgements

We are grateful for useful discussions with M. Molina Ruiz, P. Gandit, N. Trivedi, E. Shimshoni and A. Kapitulnik, and for technical support from E. André, P. Brosse-Marron and A. Gérardin. We acknowledge support from the Laboratoire dexcellence LANEF in Grenoble (ANR-10-LABX-51-01). A.F. acknowledges support from the EU project MicroKelvin, and by the US-Israel Binational Science Foundation grant no. 2014325. A.A. acknowledges support from the US-Israel Binational Science Foundation grant number 2012233 and from the Israel Science Foundation, grant number 1111/16.

## Author contributions

S.P., T.N.-D, A.F. and O.B. carried out the experiments. A.F and O.B. initiated and supervised the research. A.A. and N.D. carried out the theoretical analysis. All the authors discussed the results and wrote the manuscript.

## Additional information

**Competing financial interests:** The authors declare no competing financial interests.

**Publisher's note**: 

