## [Peer Review File · Nature Communications]

Reviewers' comments:

Reviewer #1 (Remarks to the Author):

A. In their manuscript "Quantum criticality at the superconductor to insulator transition revealed by specific heat measurements," Poran et al present experimental evidence of specific heat signatures related to quantum criticality of ultra-thin granular Pb films. They measure specific heat using a suspended platform nano-calorimeter with predeposited contact pads for simultaneous transport measurements and a metal mask to limit the deposition area for quench-condensed film growth. The transport data show a thickness-driven SIT consistent with previous measurements of granular Pb films. Heat capacity data, which can be acquired even when films are not yet conductive, show a jump at $\sim 7.2\text{K}$ for all films. This jump is coincident with the bulk T_c of Pb and demonstrates the sensitivity of the measurement to superconductivity in the grains even before the network of grains conducts, and well before the films superconduct. Surprisingly, the specific heat (C/m) at low temperatures (below T_c of the grains) increases as the film thickness decreases. This excess specific heat is the primary focus of the paper and is attributed to an increased electron effective mass due to proximity to a quantum critical point (QCP).

B. The measurement of specific heat in ultra-thin films near an SIT is a significant experimental advance because of the difficulty of the measurement undertaken, the novelty of the experiment, and the fundamental significance of specific heat to thermodynamics. However, the significance of the results to quantum phase transitions relies on the interpretation of the excess specific heat measured in films near the QCP. If the authors' claims are justified (see E), the results are likely to be of broad interest.

C. The experimental approach is valid, and the quality of the data is good. The quality of the presentation needs improvement (detailed below).

D. The signal to noise is sufficient to interpret the results. The measurement technique was previously well characterized (Ref. 19).

E. The authors make a theoretical argument for tying the excess specific heat measured to the QCP itself, however notational and reference errors and oversights make their logic very difficult to follow. I am especially concerned that the signatures they associate with the QCP change monotonically with thickness through the transition (c_s/T , α , and γ_0). If an increase of electron effective mass is expected in the vicinity of the QCP, I would expect the increase to be observed only in the vicinity of the QCP. The data show the largest effects in the thinnest films, not the films closest to the QCP.

For these reasons, I do not recommend the manuscript for publication in its present state. Below I outline suggestions for improving the clarity of the theoretical arguments as well as other minor corrections. I would support reconsidering this paper for publication in Nature Communications after revision.

F. Comments and suggested improvements:

*Overall-references to figures are missing (tex errors) in many places. The notation used for the figure and theoretical discussions (C , C_p , c_s , Δc , c_{norm}) is not consistent and makes it very difficult to follow along.

*Theoretical discussion beginning with paragraph 14- It would help to reorganize and possibly expand this section. This discussion begins by recalling the formula for specific heat of electrons in the normal state but apparently applying it to $T < T_{\text{cmf}}$. Later, c_{norm} is related to Δc_{BCS} , but it would help to reorganize this argument. The statement about Anderson localization also seems out of place (and should have a reference).

It would be very helpful to expand the discussion of the renormalization of electron mass via γ . The paper states "the experimental data indicates that $\gamma^*(T)$ significantly increases or perhaps diverges at low temperatures." It would be helpful to support this statement either by plotting an approximate γ^* or by writing out the proportionality directly in terms of measured quantities. Finally, there is no reason given that γ^* should follow a power law. Is this simply an observation? More discussion with reference to figure 3 would help.

Most importantly in this section, the phrase "toward the QCP" is used frequently when the data are shown "through the QCP." The first phrase is misleading because it implies that the observed specific heat enhancement is greatest at the QCP while the data show the greatest effects in the thinnest films.

In the discussion of the bosonic contribution to the specific heat, the citation given appears to be incorrect (Rancon & Dupuis PRE 88 2013 only includes pressure relations). Additionally, the supplemental material provided is not satisfactory, potentially because of errors referencing the supplemental figures and a missing equation. Considering the importance of this topic, a more complete discussion seems warranted.

*Minor corrections:

Abstract - the reference to the Higgs mode as equivalent to the order parameter amplitude is overly simple and potentially confusing. It is also of no relevance to the remainder of the paper so I'd suggest removing the reference here.

2nd P - excitations misspelled

4th P - "deviation from Fermi-liquid theory behavior" citation missing

7/8th P - Figure reference is missing

9th P - "critical temperature" should be "critical thickness"

10th P - Here, it would be very helpful to introduce notation: c_p , c , c_s , etc.

11th P - reminder  remainder

12th P - now uses c_p instead of c_s .

13th P - $T < T_{\text{cmf}}$ is sufficient.

Reference figure for: "The specific heat jump Δc_p also increases..."

The meaning of "smeared in amplitude" is unclear. Especially without a figure reference.

Claim: " Δc_p is more pronounced as the film is made thinner" this is the opposite of what is evident for C in Figure 1b, where we can see the jump. Perhaps reference the exact figure in supp mat, which shows Δc vs. mass.

16th P - should read: "have gapped Bogoliubov excitations." Fig reference missing

18th P - should read: "towards the QCP in a granular.."

references: Reference 3 is missing an author. Ref. 4 seems to be incorrect.

Captions/Figures: please make all notation consistent. Fig. 2c refers to c_s in the caption but the figure is labeled with " c/T ". It would also be helpful to have more annotation on the figures themselves so that the reader can more quickly identify features referred to in the text (such as Δc_p).

The supplemental information also has a number of errors and a missing equation. The captions do not seem to match the figures. This makes it difficult to follow.

G. References to previous work are mostly good. However, the references used to introduce the field (paragraph 3) are somewhat limited and only 3 of them are from the last decade of research, during which time the field has been quite active. Reference 4 also appears to be incorrect.

H. The motivation and importance of the work are clearly communicated. The manuscript becomes difficult to follow after the introduction of notation which isn't consistent throughout the paper and supplemental material. As noted above, the arguments in the theoretical discussion could be significantly improved and currently limit the interpretation of the significance of the results. Other minor comments for improvement are given in F.

Reviewer #2 (Remarks to the Author):

In this manuscript, the authors discuss a measurement of the heat capacity of granular Pb films using a custom-made cryostat with an evaporation cell for Pb and an ultra-high sensitivity calorimetry setup based on Si membrane technology. The technical aspects of this study are impressive. By consecutive deposition of thin layers of Pb in >10 incremental steps, they have tuned the Pb film from the insulating to a bulk superconducting state.

The authors present some convincing results that clearly warrant publication in a journal. For example, they observe a jump in the heat capacity at a superconducting transition temperature T_C that is roughly independent of the thickness of the film, giving strong evidence that the islands are locally superconducting long before the global phase is locked in the bulk superconducting state.

By subtracting the high-temperature background (linear plus cubic term) from the heat capacity, the authors also show that the amplitude of the jump $(C_s/T)_{\text{jump}}$ in the heat capacity at the superconducting transition T_C increases when approaching the insulating state. They interpret this as a

signature of critical fluctuations in the vicinity of a quantum critical point, enhancing the quasiparticle (QP) effective mass $m^* \sim \gamma$.

To my knowledge, there are two established methods of extracting the γ -coefficient from the heat capacity in superconducting metals:

1) By application of a magnetic field, the superconducting state can be suppressed and the heat capacity of the normal state can be measured down to the lowest temperatures. A plot of C/T vs T^2 (assuming a cubic phonon term in 3D materials) gives a zero-T intercept which can be associated with the electron mass. In the case of divergent QP mass, a modified plot of C/T vs T may be employed [1].

2) In case the former approach is not possible (high critical field H_{c2} or other experimental limitations such as no magnet in the cryostat), the heat capacity above the superconducting T_C can be fitted to the expression (1) of the main text, $C_n/T = \gamma_n + \beta T^2 + \dots$ (potentially with higher order even polynomial terms).

In the conventional approach, the jump in C/T at T_C is then compared to γ_n to yield a coupling parameter which has the numerical value 1.43 in BCS theory (see main text eq. (4)).

My main objections to the analysis in this manuscript are as follows:

- 1) The main text purports that the high-temperature heat capacity was fitted to eq. (1) and subtracted to yield the superconducting C_s ; values for β are given in table S1. However, where are the fit values for γ_n from that fit? It is crucial to compare γ_n to $(C_s/T)_{\text{jump}}$ to extract the coupling parameter, and compare the result to BCS theory. Are these films in the weak-coupling or the strong-coupling regime?
- 2) What is the t -dependence of the parameter γ_n ? At a glance, Fig 1b of the main text seems to suggest that the intercept γ_n of the large background term in the C/T vs T^2 plots actually increases with thickness t - in striking conflict with the central claim of this paper.

I have to conclude that the authors have not carefully analyzed and discussed the normal state heat capacity, and in particular γ_n . Whether this is due to experimental limitations of their setup, or due to theoretical considerations - they should at least comment on it. The amplitude of the jump in heat capacity at T_C is only an indirect indicator of QP mass; even in the most naïve (BCS) theory, it also depends on the strength of the coupling, which has not been characterized here and might have an additional t -dependence.

I am willing to consider a re-submission; but in its current state, I cannot recommend publication of this manuscript in Nature Communications.

References:

- [1] M. Brando et al., PRL 101, 026401 (2008)

REVIEWERS' COMMENTS:

Reviewer #1 (Remarks to the Author):

In their revised manuscript, Poran et al. have significantly clarified their presentation of specific heat measurements of thin granular Pb films close to the superconductor-insulator QPT. Their experimental and theoretical arguments are much more straightforward to follow than in the previous manuscript.

Most significantly, they have clarified the experimental limitations that prevent them from observing a decrease in specific heat on the low-thickness/low-mass side of the SIT. The data shown in Figure 3, top panel, are convincing. The way that precision in the specific heat measurements degrades with decreased thickness would be even further clarified by including more plots in Fig. 2c. Those chosen for that figure imply a monotonic increase in Δc_p for films as films become electronically insulating (thus on the insulating side of the SIT). The authors may also consider including error bars for γ_0 and α in Fig. 3, lower panel, which also imply a monotonic increase persisting into the insulating phase.

Additionally, the interpretation of the increase in Δc_p as an electronic effective mass increase related to superconductivity is now much easier to follow and convincing. The supplementary material is much improved and quite thorough.

One remaining question about something I overlooked the first time: I am confused about the critical thickness of these Pb films. Previously, the critical thickness of QC granular Pb films was reported to be $\sim 3.5\text{nm}$ [Jaeger, Haviland, Orr, Goldman, PRB 40 1989] and here it is closer to 11nm. Why such a large change? I started looking at critical thickness to see if there was another material that would give you better signal/noise close to the critical point. In the above reference, granular Al films had the largest thickness at the transition, but perhaps the mass/specific heat would still be low compared to Pb.

Overall, I believe the authors have addressed all of my concerns, and if they can address my relatively simple question about critical thickness, I can gladly recommend their manuscript for publication in Nature Communications.

Minor corrections/comments:

- 1) I am slightly confused by the discrepancy between the experimental cartoon and its description in the paper. 10 arms/contacts are shown in the cartoon, but 12 are referred to in the paper.
- 2) Fig 1c is labeled c_{se}/T but the axis is labeled c_{se} .
- 3) In the last paragraph on P3, I think "film" should be plural.
- 4) I still think the term "smeared" is difficult to interpret, but now the figure reference makes it clearer. "Broadened" might be better.
- 5) Fig. 3 caption uses "similar" when what is meant is "the same as"
- 6) In the conclusion, I believe "awaits" is meant instead of "awaited."

A final point that is more of an opinion: my previous comment about representing the field in the last decade was addressed partly by a mass citation on the first page of the manuscript: [7-29]. While this is definitely more inclusive, it probably doesn't help the readers and is not really the type of solution I meant to suggest. I hope the authors will either break this up or drop some of the citations that they don't feel are helpful to readers.

Reviewer #2 (Remarks to the Author):

The authors have addressed my key concerns; I understand the experimental limitations and appreciate the authors' detailed discussion in the new supplementary information, but fundamentally an analysis of the effective mass based solely on the jump in the heat capacity at TC remains somewhat unsatisfactory. In light of the remarkable experimental achievement presented here I am nevertheless willing to accept this manuscript for publication.

Some final remarks (edits):

- 1) In the Supplementary Information, could the authors add one sentence on how the mass of the film was determined in their experiment? The mass is needed when calculating the specific heat (heat capacity per volume or mass), so that its precise measurement is crucial for the interpretation of the results.
- 2) Supplementary, page 6 "we expect it to hold also for a disordered system": a reference or a sub-clause providing motivation would be welcome.

Reviewer #1

The reviewer:

The authors make a theoretical argument for tying the excess specific heat measured to the QCP itself, however notational and reference errors and oversights make their logic very difficult to follow. I am especially concerned that the signatures they associate with the QCP change monotonically with thickness through the transition (c_s/T , α , and γ_0). If an increase of electron effective mass is expected in the vicinity of the QCP, I would expect the increase to be observed only in the vicinity of the QCP. The data show the largest effects in the thinnest films, not the films closest to the QCP.

Our response:

The referee raised the problem of various notations and reference errors. In order to make the explanation clearer, the notation has been homogenized and simplified over the article. All errors in references have been corrected.

Regarding the increase in the vicinity of the QCP, we indeed expect that deep in the insulating limit, the excess specific heat should decrease. Demonstrating a decrease of specific heat in this regime is extremely difficult for two main reasons: First, pushing the sample into the insulating regime requires increasingly thinner films. This results in the signal-to-noise ratio becoming less and less favorable. Second, we cannot enter too deeply into the insulating regime because this would require measurements of a very low-mass sample beyond the sensitivity of our setup (which is state-of-the-art for specific heat experiments). We tried to improve the signal-to-noise ratio by performing new experiments (performed by Tuyen Nguyen-Duc, who is now a co-author of the work), but unfortunately we could not get better results than the ones presented here. We agree that the experiment should be improved in terms of noise, but this is apparently beyond the actual state-of-the-art in ultra-sensitive nanocalorimetry. This is now explicit in the current version (pg. 4 left column and in the conclusion).

To make this crucial point clearer, we have added a plot in the Figure 3 of the main text, showing the error bars when extracting the specific heat jumps at T_c and identifying the precise location of the QCP on the graph with a hatched area. Consequently, Figure S2 of the SM section, being now redundant, has been removed.

The reviewer:

Overall-references to figures are missing (tex errors) in many places. The notation used for the figure and theoretical discussions (C , C_p , c_s , Δc , c_{norm}) is not consistent and makes it very difficult to follow along.

Our response:

We have revised the manuscript main text and the supplementary section so that now all notations are clear, well defined and consistent. The references to the figures are now all intact.

The reviewer:

Theoretical discussion beginning with paragraph 14- It would help to reorganize and possibly expand this section. This discussion begins by recalling the formula for specific heat of electrons in the normal state but apparently applying it to $T < T_{cmf}$. Later, c_{norm} is related to Δc_{BCS} , but it would help to reorganize this argument. The statement about Anderson localization also seems out of place (and should have a reference).

Our response:

We have reorganized the theoretical section. We trust it is clearer in the current version. We made sure to explain that it is the same γ_n that determines the specific heat in both

the normal and the superconducting states. We also reorganized the statement about Anderson localization and we believe this it is now easier to understand. We added reference 46 to support our claim.

The Reviewer:

It would be very helpful to expand the discussion of the renormalization of electron mass via gamma. The paper states "the experimental data indicates that $\gamma^*(T)$ significantly increases or perhaps diverges at low temperatures." It would be helpful to support this statement either by plotting an approximate gamma* or by writing out the proportionality directly in terms of measured quantities. Finally, there is no reason given that gamma* should follow a power law. Is this simply an observation? More discussion with reference to figure 3 would help.

Our response:

The power-law function $\gamma^*(T)$ is just an experimental fit. There is no theory to support this function. It yields a good fit to the experimental results. This point is now better explained in the manuscript. We also revised figure 3 to include $\gamma^*(T)$ in the plot.

The Reviewer:

Most importantly in this section, the phrase "toward the QCP" is used frequently when the data are shown "through the QCP." The first phrase is misleading because it implies that the observed specific heat enhancement is greatest at the QCP while the data show the greatest effects in the thinnest films.

Our response:

As discussed earlier, from our results it is not possible to tell whether the specific heat decreases as the sample is pushed deep into the insulating phase. The data become too noisy to draw any definite conclusion since the film becomes too thin. This point is demonstrated clearly now in the plot of the specific heat jump at T_c (Δc_p) as a function of thickness of figure 3 (top panel). Here it is difficult to tell if the jump amplitude decreases as the QCP is crossed or not. Therefore we can only focus on the trend in the superconducting side as the sample is driven "towards the QCP" and not beyond. This is explained explicitly in the current manuscript (pg. 4 left column and in the conclusion).

The Reviewer:

In the discussion of the bosonic contribution to the specific heat, the citation given appears to be incorrect (Rançon & Dupuis PRE 88 2013 only includes pressure relations). Additionally, the supplemental material provided is not satisfactory, potentially because of errors referencing the supplemental figures and a missing equation. Considering the importance of this topic, a more complete discussion seems warranted.

Our response:

The reference has been corrected. The discussion has been expended to clarify the way the specific heat is calculated from the universal scaling function F as presented in the paper by A. Rançon, O. Kodio, P. Lecheminant and N. Dupuis, Phys. Rev. E 88, 012113 (2013). Caption and reference to figures have been corrected.

The Reviewer:

*Minor corrections:

Abstract - the reference to the Higgs mode as equivalent to the order parameter amplitude is overly simple and potentially confusing. It is also of no relevance to the remainder of the paper so I'd suggest removing the reference here.

2nd P - excitations misspelled

4th P - "deviation from Fermi-liquid theory behavior" citation missing

7/8th P - Figure reference is missing

9th P - "critical temperature" should be "critical thickness"

10th P - Here, it would be very helpful to introduce notation: cp, c, cs, etc.

11th P - reminder remainder

12th P - now uses cp instead of cs.

13th P - $T < T_{cmf}$ is sufficient.

Reference figure for: "The specific heat jump Δc_p also increases..."

The meaning of "smeared in amplitude" is unclear. Especially without a figure reference.

Claim: " Δc_p is more pronounced as the film is made thinner" this is the opposite of what is evident for C in Figure 1b, where we can see the jump. Perhaps reference the exact figure in supp mat, which shows Δc vs. mass.

16th P - should read: "have gapped Bogoliubov excitations." Fig reference missing

18th P - should read: "towards the QCP in a granular.."

references: Reference 3 is missing an author. Ref. 4 seems to be incorrect.

Captions/Figures: please make all notation consistent. Fig. 2c refers to cs in the caption but the figure is labeled with "c/T". It would also be helpful to have more annotation on the figures themselves so that the reader can more quickly identify features referred to in the text (such as Δc_p).

Our response:

We thank the reviewer for his careful evaluation of our manuscript. We have corrected all typos and mistakes and fixed all references except for the remark on P. 9 in which we argue that it should be "critical temperature" rather than "critical thickness".

The Reviewer:

The supplemental information also has a number of errors and a missing equation. The captions do not seem to match the figures. This makes it difficult to follow.

Our response:

We thank the reviewer once more. The supplemental information has been edited so that all figures and equations are correct, in place and well referenced.

The Reviewer:

References to previous work are mostly good. However, the references used to introduce the field (paragraph 3) are somewhat limited and only 3 of them are from the last decade of research, during which time the field has been quite active. Reference 4 also appears to be incorrect.

Our response:

We thank the referee. We have added a comprehensive set of references (7-28) to review previous works in the field of the SIT.

The Reviewer:

The motivation and importance of the work are clearly communicated. The manuscript becomes difficult to follow after the introduction of notation which isn't consistent throughout the paper and supplemental material. As noted above, the arguments in the theoretical discussion could be significantly improved and currently limit the interpretation of the significance of the results.

Our response:

As noted above we have revised our manuscript and supplemental material to make the notations and references clear and consistent. We have also made the theoretical arguments more clear and we trust the current version is readable and understandable.

Reviewer #2

The reviewer:

To my knowledge, there are two established methods of extracting the γ -coefficient from the heat capacity in superconducting metals:

1) By application of a magnetic field, the superconducting state can be suppressed and the heat capacity of the normal state can be measured down to the lowest temperatures. A plot of C/T vs T^2 (assuming a cubic phonon term in 3D materials) gives a zero-T intercept which can be associated with the electron mass. In the case of divergent QP mass, a modified plot of C/T vs T may be employed [1].

2) In case the former approach is not possible (high critical field H_{c2} or other experimental limitations such as no magnet in the cryostat), the heat capacity above the superconducting T_C can be fitted to the expression (1) of the main text, $C_n/T = \gamma_n + \beta T^2 + \dots$ (potentially with higher order even polynomial terms).

In the conventional approach, the jump in C/T at T_C is then compared to γ_n to yield a coupling parameter which has the numerical value 1.43 in BCS theory (see main text eq. (4)).

My main objections to the analysis in this manuscript are as follows:

1) The main text purports that the high-temperature heat capacity was fitted to eq. (1) and subtracted to yield the superconducting C_s ; values for β are given in table S1. However, where are the fit values for γ_n from that fit? It is crucial to compare γ_n to $(C_s/T)_{\text{jump}}$ to extract the coupling parameter, and compare the result to BCS theory. Are these films in the weak-coupling or the strong-coupling regime?

2) What is the t -dependence of the parameter γ_n ? At a glance, Fig 1b of the main text seems to suggest that the intercept γ_n of the large background term in the C/T vs T^2 plots actually increases with thickness t - in striking conflict with the central claim of this paper.

Our response:

The reviewer is perfectly right. The two conventional protocols proposed to analyze the data are the most commonly used in regular superconductors (crystals or thick films). The situation in very thin amorphous films is actually quite different from regular superconductors. The two accepted protocols cannot be used for two reasons: (i) the critical magnetic field is far too large (above 10T in the case of very small superconducting grains) to be used to obtain the normal state specific heat down to the lowest temperature, (ii) the phononic contribution dominates the specific heat signal above T_c giving no access to the γ in the metallic state.

These two very important points raised by the referee indeed necessitated a profound explanation in the supplemental materials. We thank the referee for this comment, and have added the following paragraph to make that point very clear in the new version of the supplemental material. With this expended explanation we hope that now our data treatment is sound.

“In order to extract information on the electronic specific heat in the superconducting state, one has to remove the contribution of phonons from the overall specific heat. In regular superconductors (crystals or thick films), the electronic specific heat dominates the total signal. Traditionally, to single out the superconducting electronic contribution to the specific heat, one can apply a magnetic field larger than the critical magnetic field thus suppressing the superconducting state. In this limit, only the normal state electrons contribute to the specific heat; the electronic contribution in the superconducting state γ_{es} is then obtained by subtracting the normal state specific heat, γ_n from the superconducting specific heat γ_{s} , i.e. $\gamma_{es} = \gamma_s - \gamma_n$. Here, this protocol cannot be applied since the expected critical magnetic field for small superconducting grains is far larger than the field that can be applied in our experiment ($B_c \gg 10$ Tesla). If the critical magnetic field happens to be too high, there is a second traditional way to extract the electronic contribution, i.e. by fitting the specific heat in the normal state above T_c using Eq. (1), and then extrapolating to temperatures lower than T_c to find γ_n . Again, this cannot be applied in our case since the electronic contribution in the normal state is overwhelmed by the

phononic contribution. The consequence of this is twofold: (i) one cannot have access to γ_n from the normal state specific heat and (ii) one can only fit the normal state c_n by a cubic power law.

Hence, in order to extract the significant information contained in the superconducting specific heat in granular materials, the following protocol has been used: The subtraction of the phononic contribution to the specific heat in the normal state is done by fitting the part of the specific heat curves above T_c only by a cubic with temperature term such as $c_i(T) = \beta_i T^3$, β_i being the i^{th} evaporation. All the relevant numerical data extracted from the heat capacity measurements are gathered in the table~\ref{table1}.”

As a concluding remark, we highlight again that we cannot have an experimental access to γ_n . This is an intrinsic limit of that kind of specific heat experiment on ultra-thin films. Even an improvement of sensitivity would not solve this limitation. Finally, regarding strong versus weak coupling, this indeed an important point in Pb materials. However, since we are comparing the same materials (Pb), the influence of the coupling should not play a role as the superconducting grains keep the same critical temperature over the entire measurement range in thickness. We do not expect that the strength of the superconducting coupling in each Pb grains is changing as a function of thickness since the size of grain is not changing.

This letter includes a point-by-point response to the reviewers' comments and detailed information about the changes made in the manuscript.

REVIEWER #1

The Reviewer:

One remaining question about something I overlooked the first time: I am confused about the critical thickness of these Pb films. Previously, the critical thickness of QC granular Pb films was reported to be ~3.5nm [Jaeger, Haviland, Orr, Goldman, PRB 40 1989] and here it is closer to 11nm. Why such a large change? I started looking at critical thickness to see if there was another material that would give you better signal/noise close to the critical point. In the above reference, granular Al films had the largest thickness at the transition, but perhaps the mass/specific heat would still be low compared to Pb.

Our Response:

Indeed the quoted paper reports on a critical thickness of ~3.5nm. However most works on quench condensed granular Pb, including our own vast experience on these systems, find a critical thickness of 9-10nm. See for example *Barber and Glover PRB 42 6754 (1990)* and *Barber and Dynes PRB 48, 10618(R) (1993)*.

The reason for this discrepancy could be the different experimental setups used. While in the Goldman group experiment the deposition sources were Knudsen cells, and, more importantly, the substrates were made of alumina, in other experiments (including ours) the films were thermally evaporated on to silicon oxide based substrates. The wetting of the quench condensed Pb is very sensitive to the substrate. Hence, different substrates may well lead to different critical thicknesses.

The Reviewer:

I am slightly confused by the discrepancy between the experimental cartoon and its description in the paper. 10 arms/contacts are shown in the cartoon, but 12 are referred to in the paper.

Our Response:

We thank the referee for pointing out this confusion. We used 10 arms for the electric contacts. This is now corrected in the revised text and Methods.

The Reviewer:

Fig 1c is labeled cse/T but the axis is labeled cse.

Our Response:

We have corrected the figure caption and it now reads c_se.

The Reviewer:

In the last paragraph on P3, I think "film" should be plural.

Our Response:

Thank you. We have corrected this typo.

The Reviewer:

I still think the term "smeared" is difficult to interpret, but now the figure reference makes it clearer. "Broadened" might be better.

Our Response:

We have used the term "broadened" instead of "smeared".

The Reviewer:

Fig. 3 caption uses "similar" when what is meant is "the same as"

Our Response:

Thank you. We have changed the term to "the same as" rather than "similar".

The Reviewer:

In the conclusion, I believe "awaits" is meant instead of "awaited."

Our Response:

The reviewer is right. We corrected this.

The Reviewer:

A final point that is more of an opinion: my previous comment about representing the field in the last decade was addressed partly by a mass citation on the first page of the manuscript: [7-29]. While this is definitely more inclusive, it probably doesn't help the

readers and is not really the type of solution I meant to suggest. I hope the authors will either break this up or drop some of the citations that they don't feel are helpful to readers.

Our Response:

We have now broken up the references representing the field of the SIT (references [7-30]) so that are grouped by the tuning parameter which is used to drive the transition (disorder, magnetic field, disorder, composition and electric field).

REVIEWER #2

The Reviewer:

...fundamentally an analysis of the effective mass based solely on the jump in the heat capacity at TC remains somewhat unsatisfactory. In light of the remarkable experimental achievement presented here I am nevertheless willing to accept this manuscript for publication.

Our Response:

We agree with the referee that the scenario presented in the discussion section based on enhancement of the effective mass is not conclusively demonstrated in our manuscript. Therefore in the revised version we pointed out that this model is a possibility to explain the way quantum criticality manifests itself in the specific heat of granular superconducting films. We slightly modified the abstract, the summary paragraph of the introduction (last paragraph of page 1), the first paragraph of the discussion (page 4) and the conclusion (page 5) accordingly.

The Reviewer:

In the Supplementary Information, could the authors add one sentence on how the mass of the film was determined in their experiment? The mass is needed when calculating the specific heat (heat capacity per volume or mass), so that its precise measurement is crucial for the interpretation of the results.

Our response:

In the Methods we explained that the thickness is determined by the well-known relationship between thickness and sheet resistance allowing to extract the mass. We confirm these numbers by the perfect agreement with the expected linear increase of the heat capacity as function of mass/thickness.

The Reviewer:

Supplementary, page 6 “we expect it to hold also for a disordered system”: a reference or a sub-clause providing motivation would be welcome.

Our response:

We have added reference [55] which demonstrates that disorder has only a quantitative effect on the collective modes in the vicinity of the SIT.

LIST OF CHANGES MADE IN THE REVISED MANUSCRIPT

1. We shortened the abstract in accordance to the Nature communications policy
2. We added sections and subsections.
3. Following the editors recommendations we transferred the supplementary information to the Methods section.
4. We changed figures 1 and 3 so that each panel is now labeled. We modified the figure captions accordingly.
5. Following Reviewer 1's suggestion we broke up references [7-30].
6. We added reference [55].
7. In the Methods we added an explanation on how the mass of the different layers was extracted.
8. We slightly modified the summery paragraph of the introduction (last paragraph of page 1), the first paragraph of the discussion (page 4) and the conclusion (page 5) in accordance with Reviewer 2's comment.
9. We corrected all the mistakes pointed out by Reviewer 1 and the editor.
10. We performed the corrections required by the editor in order to comply with the Nature communication policy.